**Data Availability Statement:** All relevant data are within the manuscript and its Supporting Information files.

# Satisfaction with regular hospital foodservices and associated factors among adult patients in Wolaita zone, Ethiopia: A facility-based cross-sectional study

**Meskerem Teka[1], Gargi Dihar[2], Tadele Dana[2], Gedion Asnake[3], Negash Wakgari[4], Zeleke Bonger[5], Wakgari Binu Daga** 🄳 [2] *

1 World Vision, Health and Nutrition Coordinator, Addis Ababa, Ethiopia, 2 School of Public Health, College of Health Sciences and Medicine, Wolaita Sodo University, Wolaita Sodo, Ethiopia, 3 School of Midwifery, College of Health Sciences and Medicine, Wolaita Sodo University, Wolaita Sodo, Ethiopia, 4 Department of Midwifery, College of Medicine and Health Sciences, Ambo University, Ambo, Ethiopia, 5 World Vision, Community Based Acute Malnutrition Office, Addis Ababa, Ethiopia

* nubonsa@gmail.com, Wakgari.binu@wsu.edu.et

## Abstract

### Background

Food service in hospital is one of the essential parts of the treatment process that determines recovery length and a hospital stay of patients. Even though many researches have been conducted on patients' satisfaction with healthcare services, there is a lack of studies that specifically address the satisfaction with food service at healthcare facilities in Ethiopia. This study aimed to assess patient satisfaction with regular hospital food service and associated factors among adults admitted to in-patient departments of hospitals.

### Methods

A hospital-based cross-sectional study design was conducted to interview 423 patients admitted to three randomly selected hospitals namely Wolaita Sodo University Referral and Teaching Hospital, Dubo St. Catholic Hospital and Sodo Christian Hospital. Participants were recruited based on probability proportional to the number of clients in each hospital. After data entry using EpiInfo v7.2.2.6, the data were exported to SPSS v23 software for further analysis. Bivariate and multivariate logistic regressions were undertaken to see the association between variables. Statistically significant variables were declared using an adjusted odds ratio with a 95% confidence interval.

### Result

Among the total participants 33.6% (95%CI: [29.1, 38.3]) of patients were satisfied with regular hospital food services. Multivariate analysis revealed that residence (AOR = 2.16; 95% CI: [1.28, 3.63]), monthly income (AOR = 5.64; 95%CI: [2.30, 8.28]), flavour of meal, (AOR = 2.63; 95%CI: [1.34, 5.56]), and provision of easily chewable food (AOR = 7.50; 95%CI: [2.00, 12.82]) were influencing factors for satisfaction on hospital foodservices.

**Funding:** The author(s) received no specific funding for this work.

**Competing interests:** The authors have declared that no competing interests exist.

## Conclusion

This research ascertained a low patient satisfaction with regular hospital meal service. The identified factors need to be addressed giving attention for each foodservice dimension to scale up the patient satisfaction with hospital food services.

## Introduction

### Background

Patient satisfaction has become a key measure by which the quality of health care services is evaluated. Available studies had identified the overall hospital patient satisfaction, focusing on quality of nursing, physician, and the technical medical care, [1, 2]. Inpatient satisfaction is not only about core services in health care but also about communications, sociability, and food services in hospitals, [3, 4]. Foodservice in hospitals is the only source of nutrition for majority of patients and essential part of the healthcare management. It is one of the most important items of health care quality perceived by patients and by their families, [5–7]. The provision of good patient meals should be regarded as a part of hospital treatment which can promote recovery, [8–11].

The provision of hospital food service is a widespread problem in all over the world that is causing under-nutrition among patients in hospitals due to dissatisfaction towards foodservice, [12, 13]. Malnutrition at hospital is a problem that increases the severity of illnesses, the recovery duration, length of stay in the hospital, and the medical cost, [14, 15]. Evaluation of patients' satisfaction with food services in the hospitals is one of the strategies to prevent malnutrition. Available evidence identified low monthly income, patients' perception on meal temperature, servers' characteristics, mealtime, meal taste, portion size, texture, nutritional information, responsiveness to foodservice problem, menu variety and sanitation as determining factors [5, 16–19].

Studies show that satisfaction on hospital foodservices varies from country to country. The overall satisfactions towards the overall quality of foodservices were 91% in Pakistan, [20] and 78.8% in Saudi Arabia, [5]. The results of overall satisfaction in Malaysia indicated that 53.3% of patients rated hospital foodservice as okay and 32% of respondents rated hospital foodservice as either very good or good, [21]. However another study conducted in Kenya showed that 64.3% of inpatients were not satisfied with overall quality of hospital food, in which 96.9, 76.5, 71.4%, and 65.3 of the patients were not satisfied with the variety, type, taste, and appearance of hospital food respectively, [22].

In addition, some aspects could affect patient's satisfaction on hospital foodservices based on food characteristics and distributions of served meals, [7]. Accordingly, the most satisfied aspect by the patients was the cleanness of food equipment, 96.8% in Saudi Arabia, [5] and 50% of the patients were not satisfied with the time of meal serving and the delay of mealtime affects patient satisfaction and meal intake in many directions regarding meal taste, appetite, meal temperature cool or heat in Sohag University Hospital, [23].

### Overall healthcare facilities in Ethiopia

The government of Ethiopia had developed a national nutrition strategy and program (NNP) to address the nutritional issue at health facilities. Accordingly, the Ethiopian Health Sector Development Program Four (HSDP IV) had integrated the nutrition into health facilities

through a program Health Facility Nutrition Services to keep best nutritional status of patients admitted to health facilities, [24, 25]. The Ethiopian Standard Agency (ESA) had incorporated food and dietary services as one of medical services into health facility requirements. It recommended health facilities at all levels to provide nutritionally adequate meals and at least three meals per day to facilitate the recovery of patients, [26, 27].

According to the report of Ministry of Health in 2020 (2012 Ethiopian Fiscal Year) there were a total of 353 functioning public hospitals providing different medical cares in Ethiopia, [28]. Regarding the quality of healthcare services, private health facilities deliver better quality of services to their clients compared to public health facilities in Ethiopia. Available evidence showed high rate of satisfaction in all aspects of medical care among clients at private hospitals in Ethiopia, [29–31].

Knowledge of patient satisfaction on hospital foodservice is an important basis to provide quality health services over time so that the health outcomes of patients will be improved. Even though foodservice aspects are the most salient influences of health services satisfaction, the relationships between satisfaction and the hospital foodservice is not explored among inpatients in Ethiopia. This study aimed to assess patient satisfaction with food services and associated factors among adult patients admitted at hospitals.

## Methods and materials

### Setting and study design

This hospital-based cross-sectional study was conducted from October to November 2019 in Wolaita zone, southern Ethiopia. Wolaita zone is in Southern Nations, Nationalities and People's Regional State of Ethiopia at 314 km south from Addis Ababa. The estimated total population of the zone was 2,492,887 in 2019, [32]. Wolaita zone is further divided into 16 districts, 6 town administrations and 355 Kebeles (smallest administrative unit). There are 7 hospitals, 68 health centres, 352 health posts and 180 private Clinics. All hospitals are providing food service but the other health facilities not. According to the report of Wolaita Zonal Health Department in 2019, the 7 hospitals were providing healthcare services for 21,354 admitted patients. The study was conducted in three randomly selected hospitals namely Wolaita Sodo University Teaching and Referral Hospital (WSUTRH), Dubo St. Catholic Hospital and Sodo Christian Hospital. Dubo St. Catholic and Sodo Christian hospitals are private's organizations while WSUTRH is public's hospital. There were 2,893 health care professionals and 2601 supportive staff of various categories in the selected hospitals, [Wolaita Zonal Health Department fiscal year report, 2019].

### Population and sampling

All adult patients admitted to hospitals in the zone were the source population. All adult patients admitted to the randomly selected hospitals were the study population while the consecutively selected patients were the study participants. Eligible participants were adult inpatients aged 18–64, capable to communicate and at least 3 nights since admitted to the hospitals. This period is thought to be enough for patients to give their experiences on hospital foodservice [22, 33]. Patients who were not receiving meals during their hospital stay, on tube feeding, unable to communicate and admitted to psychiatric wards were excluded from the study.

The sample size was calculated using a single population proportion formula considering a 95% level of confidence, a 5% margin of error, and a 10% non-response rate. Since there is no published information in Ethiopia, we have used 50% as proportion of patient satisfaction with regular hospital foodservice giving 423 final sample sizes.

A specific sample size for each selected health facility was decided using proportion-to-size allocation based on client load obtained from the quarterly report of each selected hospital. Data were collected from patients admitted to medical, surgical, orthopaedic, and gynaecologic wards in each hospital. On day of data collection, patients who fit the criteria were identified with the help of health professionals who were assigned in the respective wards. Within each hospital, patients fulfilling the inclusion criteria were consecutively enrolled until sample size was reached.

## Measurements

**Patient satisfaction with hospital food service.** Was the outcome variable of the study and measured using 14 item questions related to satisfaction towards food services adopted from a study conducted in Saudi Arabia, [5]. Patients were asked to show their level of satisfaction by selecting responses on a five-point Likert scale (rated as strongly disagree = 1, disagree = 2, neutral = 3, agree = 4 and strongly agree = 5). To see the total score of each respondent, the points obtained from the 14 items were computed. A total score ranges from 14 to70 points for each respondent in which higher scores indicate greater satisfaction. A mean score of the Likert scale was computed giving 2.44. A client was regarded as satisfied if scored mean of ≥2.44 and considered as dissatisfied if scored mean of <2.44.

**Independent variables.** Included a set of socio-demographic characteristics (age, sex, residence, marital status, religion, occupation, educational status, and monthly income), length of hospital stay and hospital foodservices dimensions (food characteristics, meal serving approach and physical environment) with their respective aspects as independent variables. Continuous variables were categorized in the way that they were appropriate for our analysis. Accordingly, Age was categorised as 18–30, 31–43, 44–56, and 57–65 years, monthly income was categorized as ≤500, 501–1000, 1001–3000 and ≥3001 in Ethiopian birr (ETB) [1 USD was equivalent to 29.55 ETB during the data collection period], length of hospital stay after admission was categorized as 3–7, 8–14 and ≥15 days.

## Data collection and quality assurance

The tool for data collection was developed after reviewing literature [18–23, 33, 35]. The structured questionnaires were prepared first in English and then translated to the local languages, Amharic, and Wolaita Donna, [**S1 Appendix**]. The questionnaire was back-translated into English by another language expert to keep consistency in the meaning of words and concepts. The contents of the questionnaire included socio-demographic variables, food characteristics, serving approach, physical environment factors and the overall patient satisfaction on hospital foodservices.

Data collection was conducted using interviewer-administered structured questionnaire in either of the local language. Data were collected by six diploma health professionals and supervised by two BSc health workers recruited based on their earlier experiences. Data collectors were fluent speakers of the two local languages who could use either of the Wolaita Dona or Amharic language for interview. Two days training was given to data collectors and supervisors to make them familiar with the aims, and the method of the research. The training aimed to enhance their interviewing skills and ensure consistent interviews to be done by data collectors.

Before the actual data collection, pre-test was done on 5% of respondents in another hospital which was not selected for the actual data. Completeness of each questionnaire was checked by the supervisors on daily basis. The overall process was watched by the principal investigator. To keep the internal consistency of the tool and to determine the reliability of the test,

Cronbach's alpha was applied. If the alpha was high ($\geq$0.70), the item was reliable and the test was considered as internally consistent. If the items in the test had a low correlation, rejecting the item that was inconsistent with the rest and retaining the item with the highest average inter-correlation was done via item analysis, [34].

## Data analysis

The collected data was coded and entered EpiData v3.1. Then the data was exported to SPSS statistical software v23 for cleaning and further analysis. The errors identified were corrected after checking the questionnaires. Summary measures such as frequencies, and the regression analysis were computed. Binary and multivariable logistic regression analysis which provided odds ratio with 95%CI was used to show statistical associations. Variance inflation factors (VIF, cut off point = 3.4) to detect multicollinearity among the independent variables were assessed before incorporating them into the multivariable logistic regression model.

Variables from binary logistic regression analysis with P-value less than 0.25 were entered to multivariable logistic regression to control the possible confounders. The statistical significance of independent predictors of satisfaction on hospital foodservice was declared using AOR with 95%CI. The Hosmer-Lemeshow goodness of fit test was done to check the ability of the model whether it can discriminate between participants who were satisfied with hospital foodservices and those who did not (P-value = 0.56).

## Ethical approval and consent to participate

Ethical approval was obtained from the Ethical Review Committee of the College of Health Sciences and Medicine at Wolaita Sodo University with Ref. No. CARD731/730/12. Formal permission was obtained from the health departments of Wolaita Zone and the respective hospitals where the study was conducted. The information about informed consent was explained to the participants in their own language. Respect for the participants (autonomy) was kept. The purpose of the research, and the expected duration for the participant to complete the interview was communicated. This study kept the confidentiality of the participants; their name was not written on any result (anonymity was ensured). Participants had the right to withdraw or to interrupt their participation at any time without penalty or loss of benefits. We obtained the informed verbal consent from each of the participants.

## Results

### Socio-demographic characteristics of respondents

Among 423 participants of the study, 207(48.9%) were male. The mean age of the respondent was 41 [SD: 8.72] years. The majority of the study participants, 221(52.2%) were from urban and 290 (68.6%) were married. One hundred thirty-two (31.2%) of the participants had no formal education, eighty-seven (20.6%) of participants attended higher education. One hundred twenty-one (28.6%) respondents were house-wives. One hundred four (24.6%) reported a monthly income of less than 500 Ethiopian birr (ETB) and 123(29.1%) were getting monthly income 1001–3000 ETB, [Table 1].

### Characteristics of foodservices in hospitals

In this study, participants had reported their experiences on the goodwill part of foodservices for few characteristics of foods. Accordingly, 417(98.6%), 386(91.3%), and 399(94.3%) of the patients reported they were receiving three meals daily, well-cooked meals for easily chewing, and choose among two or more varieties of food in a meal respectively. However, a

**Table 1. Socio-demography characteristics of hospitals admitted adult patients at hospitals, 2019.**

| Characteristics | | Frequency | Percentage |
|---|---|---|---|
| Age distribution (in years) | 18–30 | 64 | 15.1 |
| | 31–43 | 203 | 48.0 |
| | 44–56 | 143 | 33.8 |
| | 57–65 | 15 | 3.6 |
| Sex | Male | 207 | 48.9 |
| | Female | 216 | 51.1 |
| Residence | Urban | 221 | 52.2 |
| | Rural | 202 | 47.8 |
| Marital status | Single | 65 | 15.4 |
| | Married | 290 | 68.6 |
| | Divorced | 31 | 7.3 |
| | Widowed | 37 | 8.7 |
| Religion | Orthodox | 78 | 18.4 |
| | Protestant | 268 | 63.4 |
| | Muslim | 56 | 13.2 |
| | Catholic | 21 | 5 |
| Occupation | Government employee | 119 | 28.1 |
| | Merchant | 106 | 25.1 |
| | Farmer | 77 | 18.2 |
| | Housewife | 121 | 28.6 |
| | Others (daily labourer, NGOs) | 6 | 1.4 |
| Educational Status | No formal education | 132 | 31.2 |
| | Primary school completed | 122 | 28.8 |
| | Secondary school completed | 82 | 19.4 |
| | Higher education and above | 87 | 20.6 |
| Monthly income in ETB* | ≤500 | 104 | 24.6 |
| | 501–1000 | 122 | 28.8 |
| | 1001–3000 | 123 | 29.1 |
| | >3001 | 74 | 17.5 |
| Length of hospital stay (in days) | 3–7 | 38 | 9 |
| | 8–14 | 217 | 51.3 |
| | ≥15 | 168 | 39.7 |

*1 USD was equivalent to 29.55 ETB during data collection period

considerable proportion of study participants had unacceptable experiences on the majority of the food characteristics. So, 423(100%), 399(94.3%), and 230(54.4%) of the participants reported no provision of water with their meal, unacceptable flavour of the meal to eat, and avoided the meal because of taste aversion respectively, [**Fig 1**].

## Patient's satisfaction level with regular hospital foodservices

Patients were asked for their degree of agreement on 14 foodservices items and level of satisfaction was assigned either as satisfied or dis-satisfied based on the mean score. The overall satisfaction towards hospital foodservices among the 423 participants was 33.6% (95%CI: [29.1, 38.3]). Regarding the 14 items assessing satisfaction level, most of participants disagreed/ strongly disagree on majority of the items. So, patients disagreed on general cleanness of room (92.7%), meal appearance (90.5%), room privacy (89.4%), flexibility of bed (87.9%), number of

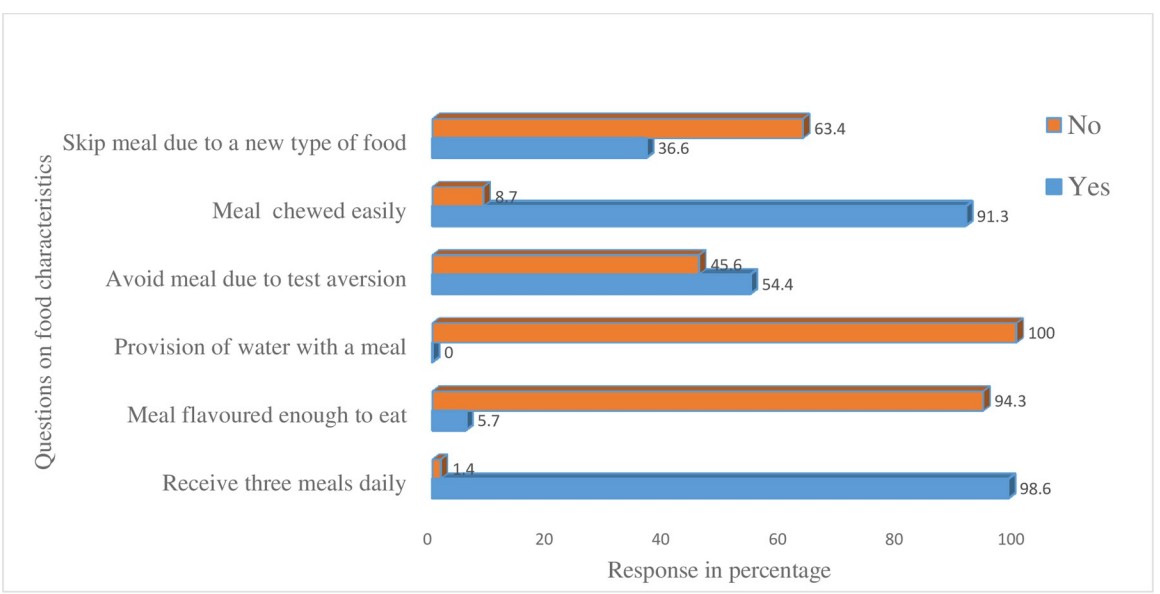

**Fig 1. Characteristics of hospital foods among patients in hospitals, Wolaita zone, Ethiopia 2019.**

meals served per day (86.8%), taste of meal (85.1%), amount of meal (75.2%) and warmness of the served meal (67.1%). Participants also strongly disagreed on variety of meal (70.9%) and goodness of meal to health (64.3%). However, patients agreed on timely distribution of meal (70.9%), staff behaviour (68.1%), cleanliness of plates (65.2%) and cleanliness of cups (64.8%), [**Table 2**].

## Factors affecting the patient's satisfaction level

During the bivariate logistic regression analysis variables like residence, sex, monthly income, length of hospital stay, skiping meal due to new typeof meal, three meal per day, easily

**Table 2. Satisfaction level with hospital foodservices among adult patients, 2019.**

| Satisfaction level | Strongly disagree N (%) | Disagree N (%) | Neutral N (%) | Agree N (%) | Strongly agree N (%) |
|---|---|---|---|---|---|
| The meal served is tasty | 12(2.8) | 360(85.1) | 0 | 51(12.1) | 0 |
| The meal appearance is good | 0 | 383(90.5) | 0 | 40(9.5) | 0 |
| The variety of meal is enough | 300(70.9) | 93(22.0) | 8(1.9) | 22(5.2) | 0 |
| The warmness of served meal is good | 6(1.4) | 284(67.1) | 1(0.2) | 132(31.2) | 0 |
| Meal distribution is timely | 28(6.6) | 90(21.3) | 5(1.2) | 300(70.9) | 0 |
| Amount of meal is enough | 2(0.5) | 318(75.2) | 1(0.2) | 102(24.1) | 0 |
| Plates are clean | 20(4.7) | 118(27.9) | 9(2.1) | 276(65.2) | 0 |
| Cups are clean | 18(4.3) | 129(30.5) | 2(0.5) | 274(64.8) | 0 |
| Staff behaviour is good | 10(2.4) | 118(27.9) | 7(1.7) | 288(68.1) | 0 |
| Number of meals served per day is enough | 0 | 367(86.8) | 0 | 56(13.2) | 0 |
| Meal served is good to health | 272(64.3) | 130(30.7) | 1(0.2) | 20(4.7) | 0 |
| General cleanness of room is good | 5(1.2) | 392(92.7) | 12(2.8) | 14(3.3) | 0 |
| Bed flexibility is comfortable to eat | 38(9.0) | 372(87.9) | 8(1.9) | 5(1.2) | 0 |
| Level of room privacy is good | 32(7.6) | 378(89.4) | 6(1.4) | 7(1.7) | 0 |
| **Overall patient satisfaction towards hospital foodservices** | **Satisfied** | | | **142(33.6)** | |
| | **Dissatisfied** | | | **281(66.4)** | |

**Table 3. Bivariate and multivariate logistic regression analysis of patient satisfaction with foodservice among adult patients, 2019.**

| Variables | | Satisfaction n (%) | | COR (95%CI) | AOR (95%CI) |
|---|---|---|---|---|---|
| | | Satisfied | Dissatisfied | | |
| Residence | Urban | 51(26.4) | 142(73.6) | 1 | 1 |
| | Rural | 91(39.6) | 139(60.4) | 1.82(1.06–2.39) | **2.16(1.28–3.63)**\* |
| Sex | Male | 38(66.7) | 19(33.3) | 5.04(2.78–9.14) | 1.87(0.78–4.46) |
| | Female | 104(28.4) | 262(71.6) | 1 | 1 |
| Income (ETB) | ≤500 | 27(67.5) | 13(32.5) | 4.84(2.411–9.72) | **5.64(2.30–8.28)**\* |
| | >500 | 115(30) | 268(70) | 1 | 1 |
| Hospital stay (in days) | 3–14 | 66(38.4) | 106(61.6) | 1.43(.95–2.16) | 1.06(.06–1.76) |
| | >14 | 76(30.3) | 175(69.7) | 1 | 1 |
| Meals have good flavour | Yes | 76(39.4) | 117(60.60) | 1.61(1.53–17.17) | **2.63(1.34–5.56)**\* |
| | No | 66(28.7) | 164(71.3) | 1 | 1 |
| Skip meals b/c of new meal type | Yes | 6(66.7) | 3(33.3) | 4.09(1.01–16.6) | 1.70(0.75–3.88) |
| | No | 136(32.9) | 278(67.1) | 1 | 1 |
| Three meals daily | Yes | 10(71.4) | 4(28.6) | 5.25(1.62–17.04) | 0.45(0.03–7.74) |
| | No | 132(32.3) | 277(67.7) | 1 | 1 |
| Well-cooked to Chew easily | Yes | 26(76.5) | 8(23.5) | 7.65(3.36–17.40) | **7.50(2.00–12.82)**\* |
| | No | 116(29.8) | 273(70.2) | 1 | 1 |
| Bed position flexible to eat | Yes | 11(57.9) | 8(42.1) | 2.87(1.13–7.293) | 1.95(0.70–5.47) |
| | No | 131(32.4) | 273(67.6) | 1 | 1 |
| Hand washing facility | Yes | 17(94.4) | 1(5.6) | 3.08(5.01–29.30) | 9.76(0.95–19.93) |
| | No | 125(30.9) | 280(69.1) | 1 | 1 |
| Need help to eat | Yes | 139(35.5) | 253(64.5) | 5.13(1.53–17.17) | 4.12(0.45–7.88) |
| | No | 3(9.7) | 28(90.3) | 1 | 1 |

\* Statistically significant in multivariable logistic regression (AOR) at p<0.05

chewable food, need help to eat, meal flavored and provision of freshwater were statistically associated with patient satisfaction towards hospital foodservices. However, educational status, marital status, age and occupation were not associated.

In the multivariable logistic regression analysis, residence, monthly income, meal flavor, and well-cooked food easily chewable food were statistically significant whereas sex, length of stay, need help to eat the served food, skip meal because of a new type of meal, get three meal per day were not statistically significant, [Table 3]. Participants who were from rural residence were about 2 times more likely to be satisfied compared to those who were from urban residence (AOR = 2.16, 95%CI: [1.28, 3.63]). Participants who were getting monthly income of equal to or less than 500 Ethiopian birr were about 5.6 (AOR = 5.64, 95% CI: [2.30, 8.28]) times more likely to be satisfied compared to those who were getting monthly income of more than 500 Ethiopian birr. Participants who reported that they liked the flavour of the food were about 2.6 (AOR = 2.63, 95%CI: [1.34, 5.56]) times more likely to be satisfied compared to those who reported that they did not like the flavour. Participants who reported that the food was well-cooked to be chewed easily were 7.5 (AOR = 7.50, 95%CI: (1.99, 28.23) times more likely to be satisfied compared to those who reported that the food was not easily chewable.

## Discussion

This study assessed important information about patients' satisfaction and potential factors regarding hospital food service in wolaita zone. The result showed that 33.6% (95%CI: [29.1,38.3]) of the patients were satisfied with overall foodservices in the hospitals.This finding

was lower than findings of studies conducted in other countries like 78.8% in Saudi Arabia, 53.3% in East Malaysia, and 64.2% in Egypt, [5, 21, 23]. The possible justification for the difference might be the difference in socioeconomic, cultural, infrastructural factors between our study area and those countries.

Patients at hospitals had reported different experiences towards various aspects of foodservices. Accordingly, all (100%) and 399(94.3%) of the participants reported that there was no provision of water and the flavour of the meal was not enough to eat, respectively. In addition, (97.9%) and 369(87.2%) of participants were okay with the timeliness of food distribution and cleanliness of cups, respectively. These findings were supported by earlier studies conducted in other countries [5, 7, 23]. This could be explained as different dimensions and aspects of hospital foodservices were influencing patient satisfaction. This indicates the need for interventions to be made to address the influencing factors considering each dimension and aspect.

Regarding the predicting factors, patients from the rural area were more likely to be satisfied with hospital foodservice compared to patients who came from urban area. The findings from a study conducted in Egypt at Sohag University Hospital supported this result, [23]. The probable reason might be the fact that rural communities could have less access to alternative meals. In this study, patients who were getting equal or less than 500 Ethiopian birr were more likely to be satisfied than patients who were getting more than 500 Ethiopian birr per month. Other earlier studies reported similar findings, [5]. This could be explained as patients with lower income were more dependent on the food provided by hospitals due to difficulty of getting alternative and adequate food.

In addition, patients who perceived that the hospital meal have good flavour to eat were more likely to be satisfied when compared to patients who perceived that the hospital meal has not good flavour to eat. Earlier studies also suggested that improving the flavour of hospital meals with different enhancers is important to meet expectation of patients towards foodservices, [35]. This might imply that hospitals need to add more flavour to meals with fresh produce so that patients are more satisfied with foods. Concerning the characteristics of hospital foods, patients who perceived that the meal is well-cooked to chew easily were more likely to be satisfied than those who perceived that the served food is not well-cooked for easily chewable. The possible explanation of this finding might be the provision of well-cooked food improves the feeling of eating which can increase the satisfaction level of the serving food.

## Strength and limitation of the study

The strength of this study was that it was a single study conducted in Ethiopia assessing patient satisfaction and associated factors on hospital foodservices. So it might be a baseline source of information for the health managers and future studies in the country. The possible limitations of the current study were that it focused on the overall patient satisfaction from the patient perspective only. It did not assess the organizational perspectives like skills of cooking staff, the team distributing meals, health professionals and food quality of the served food. In addition, even though, identification of potential factors using odds ratio is widely acceptable, this cross-sectional study design could not detect the causal relationship between patient satisfaction on hospital foodservices and influencing factors. The result of this study should be carefully generalized for the population in other areas especially at the central part of the country, as health system in Ethiopia is better at the centre.

## Conclusion and recommendation

The findings of this research revealed low patient's satisfaction with hospital foodservice among hospital patients. In this study, a significant proportion of participants were okay with

serving approach of foodservice while the majority of them had reported unacceptable experiences on aspects of food characteristics and physical environment. Residence, monthly income, flavour of meals, and well-cooked meals were identified as statistically associated factors with patient satisfaction on foodservice at hospitals. Our results also in sighted the interventions need to be made to address the influencing factors considering each foodservice dimension and aspect. Policymakers and health managers need to work on how to maintain the good aspects of hospital foodservices like timely distribution of meals, accepted staff behaviour, flavour of meals and cooking meals. On the other hand, health managers should meet the need of clients on the general cleanness and privacy of the room.

## Lessons for international healthcare managers

Assessing patient experiences towards hospital food services and utilizing best research evidence can make a more transparent and sustainable food and dietary service, to which clients are central. Since this study is the single data in Ethiopia, healthcare managers need more data to design and implement strategies for improvement of food services at hospitals in Ethiopia.

## Supporting information

**S1 Appendix. Questionnaire in English, Amharic, and Wolaita Donna.**
(PDF)

## Acknowledgments

We would like to acknowledge Wolaita Sodo University for giving us the opportunity for this study. We would like to extend our gratitude to Wolaita Zone Health department and the respective hospitals for their all aspects of unreserved support when we needed. Our acknowledgment also goes to data collectors, supervisors, and study participants.

## Author Contributions

**Conceptualization:** Meskerem Teka, Gargi Dihar, Tadele Dana.

**Data curation:** Meskerem Teka.

**Formal analysis:** Meskerem Teka, Zeleke Bonger, Wakgari Binu Daga.

**Funding acquisition:** Meskerem Teka.

**Investigation:** Meskerem Teka.

**Methodology:** Meskerem Teka, Negash Wakgari, Wakgari Binu Daga.

**Project administration:** Meskerem Teka.

**Resources:** Meskerem Teka.

**Supervision:** Meskerem Teka, Wakgari Binu Daga.

**Validation:** Meskerem Teka, Gargi Dihar, Tadele Dana, Gedion Asnake.

**Visualization:** Meskerem Teka, Gargi Dihar, Tadele Dana, Wakgari Binu Daga.

**Writing – original draft:** Meskerem Teka.

**Writing – review & editing:** Meskerem Teka, Gargi Dihar, Tadele Dana, Gedion Asnake, Negash Wakgari, Zeleke Bonger, Wakgari Binu Daga.

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
