## [Decision Letter · Decision Letter 0]

15 Apr 2021

PONE-D-20-36652

Satisfaction with Regular Hospital Foodservices and Associated Factors among Adult Patients in Wolaita zone, Ethiopia: a facility-based cross-section study

PLOS ONE

Dear Dr. Daga,

Thank you for submitting your manuscript to PLOS ONE. After careful consideration, we feel that it has merit but does not fully meet PLOS ONE’s publication criteria as it currently stands. Therefore, we invite you to submit a revised version of the manuscript that addresses the points raised during the review process.

The manuscript has been evaluated by two reviewers, and their comments are available below. You will see the reviewers have congratulated you on the importance of your work. However, they have also raised a number of concerns that should be addressed before the manuscript can be further considered for publication.

The key concerns noted by the reviewers relate to the reporting of the study setting. Specifically, the reviewers requested additional information about the hospitals included in the study, as well as throughout Ethiopia, details about the food service, and clarity about the questionnaires and cut-off points.

We look forward to receiving your revised manuscript.

Kind regards,

Danielle Poole

Staff Editor

PLOS ONE

Journal Requirements:

Please include in your Methods section (or in Supplementary Information files) the participating hospitals/institutions.

We suggest you thoroughly copyedit your manuscript for language usage, spelling, and grammar. If you do not know anyone who can help you do this, you may wish to consider employing a professional scientific editing service. 

Reviewers' comments:

Reviewer's Responses to Questions

**Comments to the Author**

1. Is the manuscript technically sound, and do the data support the conclusions?

Reviewer #1: Yes

Reviewer #2: Yes

2. Has the statistical analysis been performed appropriately and rigorously? 

Reviewer #1: Yes

Reviewer #2: Yes

3. Have the authors made all data underlying the findings in their manuscript fully available?

Reviewer #1: Yes

Reviewer #2: Yes

4. Is the manuscript presented in an intelligible fashion and written in standard English?

Reviewer #1: Yes

Reviewer #2: No

5. Review Comments to the Author

Reviewer #1: Dear Authors

I would like to congratulate you for choosing an appropriate topic for research in healthcare setups. I have read complete manuscript. Please find below recommendations.

1) Add one separate section on “Lessons for International Healthcare Managers” at the end of manuscript.

2) Mention type of hospitals (private or public) for 03 selected hospitals in abstract and other sections of manuscript.

3) Add one separate section on “Overall Healthcare Facilities in Ethiopia”, as a sub part of introduction section, which may include,

a. number of private and public healthcare service providers in the country

b. general comparison of quality of services between private and public service providers

Regards,

(Reviewer)

Reviewer #2: 1. Summary of the research

The study title is: “Satisfaction with Regular Hospital Foodservices and Associated Factors among Adult Patients in Wolaita zone, Ethiopia: a facility-based cross-section study”. The study was conducted in 3 hospitals and established that about a third (33.6%) of the patients were satisfied with hospital food services. Factors such as the patients’ residence, monthly income, flavor of the meal, and ease of chewing the food were associated with satisfaction with regular Hospital Foodservices.

This is a unique study, because there are limited studies of this nature in sub-Sahara Africa, and this has implications on the nutrition status of hospitalized patients, as well as the negative consequences of malnutrition in hospital, such as length of stay, mortality, and cost of treatment.

2. Examples and evidence

a. MAJOR ISSUES

• The authors can consider working with an editor to improve the quality of the manuscript. There are several grammatical and sentence structure issues throughout the document that should be addressed – a few examples include lines 50-51, 55-56, 88-89, 173-174, 180-181.

Consider paying closer attention to Lines 181-187. The information conveyed is not clear due to the language, yet it is a key finding in the study that needs clear and concise communication. See highlighted area here …..“Accordingly, 417(98.6%) and 386(91.3%) of the patients reported that they were receiving three meals daily and chewable meals respectively. Study participants had unacceptable experiences on majority of the food characteristics they were asked. In relation to this, all (100%) of the participants reported that there was no provision of water with their meal, and 399 (94.3%) of patients say that the flavor of the meal was not enough to eat. In addition, significant portion of participants, 230(54.4%) and 155(36.6%) had avoided the meal because of taste aversion and had skip the meal because of new meal type respectively,……”

• Indicate the steps taken to ensure the validity of the data collection instruments as well as procedures. For example, what was the source of the questionnaires used?

• Indicate how the cut-off for the “overall patient (dis)satisfaction towards hospital foodservices” was determined (table 3). How was it decided that 142 were Satisfied, while 281 were Dissatisfied 281?

• Consider rephrasing the lines 214-217, by using interpretive language for the readers to get the results in a clearer manner, as this is a key component of the study. For instance, rephrase the section on meal flavor, easily chewable food to something like, …. “those who reported that they liked the flavor of the food were 2.6 times more likely to be satisfied compared to those who reported that they did not like the flavor …”.

THIS IS DONE IN THE DISCUSSION SECTION. Consider moving some of the content from the discussion section to the findings, so that the discussion is focused on situating this study in existing literature, as well as the implication of the findings.

• Consider merging contents of table 2 and table 3 that are presenting the same aspects of service. Currently, there is some overlap of information in the two tables. ONE SUGGESTION IS TO MAKE TABLE 2 AS CHARACTERISTICS OF SERVICE that were objectively determined e.g. number of meals, availability of water, availability of meal choice, whether they ate or skipped meals, among others. THEN IN TABLE 3, report all the subjective aspects that are the patients’ perceptions/ experiences.

• Consider giving a general description of the foodservices so that the reader has some context, e.g. cover things like whether there was a variety of foods to choose from?

• Consider doing analysis by hospital as a factor since the different hospitals may have different characteristics and experiences

b. MINOR ISSUES

• In the introduction, examples from similar settings, particularly African countries, can be included to allow for comparison for levels of dissatisfaction (lines 58 and 67).

• The introduction can be boosted to include factors associated with (dis)satisfaction now that this is part of the title of the study

• Provide/cite the references used for the population and sampling content

• Lines 86-83 can be revised to describe the sampling process in a step-by-step clearer language.

• Give a better description of the hospitals that were included in the study

• Put ETB in full first before abbreviating, in lines 176-177

• Table 1: Indicate the equivalence of Ethiopian Birr to USD for international readers

• Line 199, the meaning of “means of hygiene (95.7%)” is unclear, consider rephrasing

• Correct the spelling for “TASTE” from “TEST” in Figure 1 and in Table 3, row 2

• If information on patients’ satisfaction with food texture and/ or temperature is available, this can be added to the findings

• The authors can ensure consistency in the way concepts are presented, by avoiding the interchanging of words that can result in different meanings in the text and tables. For instance, clarify whether when you are speaking with the patients on the chewability of the food (lines 182, 215, Table 4, Figure 1), was it that the food itself was tough to chew due to undercooking, or was it that the patients themselves had chewing issues, or both? The table 4 presentation versus that in the text (line 215) may differ in meaning

• Clarify meaning of “means of hygiene” in line 193

6. PLOS authors have the option to publish the peer review history of their article (what does this mean?). If published, this will include your full peer review and any attached files.

Reviewer #1: **Yes: **Dr. Vishal Kamra

Reviewer #2: **Yes: **Irene Ogada

---

## [Author Response · Author response to Decision Letter 0]

29 Jun 2021

Manuscript ID: PONE-D-20-36652

Title: Satisfaction with Regular Hospital Foodservices and Associated Factors among Adult Patients in Wolaita zone, Ethiopia: a facility-based cross-sectional study 

Journal: PLOS ONE

Dear Editor and Reviewers,

Thank you for giving us an opportunity to revise our manuscript. We found your comments/feedback very helpful in improving the manuscript and we have revised the manuscript accordingly.

Here we tried to re-phrase our title a little bit by replacing ‘cross-section’ with “cross-sectional” as follows: 

Satisfaction with Regular Hospital Foodservices and Associated Factors among Adult Patients in Wolaita zone, Ethiopia: a facility-based cross-sectional study

Dear Dr. Daga,

Thank you for submitting your manuscript to PLOS ONE. After careful consideration, we feel that it has merit but does not fully meet PLOS ONE’s publication criteria as it currently stands. Therefore, we invite you to submit a revised version of the manuscript that addresses the points raised during the review process.

The manuscript has been evaluated by two reviewers, and their comments are available below. You will see the reviewers have congratulated you on the importance of your work. However, they have also raised a number of concerns that should be addressed before the manuscript can be further considered for publication.

The key concerns noted by the reviewers relate to the reporting of the study setting. Specifically, the reviewers requested additional information about the hospitals included in the study, as well as throughout Ethiopia, details about the food service, and clarity about the questionnaires and cut-off points.

We look forward to receiving your revised manuscript.

Kind regards,

Danielle Poole

Staff Editor

PLOS ONE

Journal Requirements:

Response: Thank you for your guidance. We have revised the manuscript in accordance with the requirement.

2. Please include in your Methods section (or in Supplementary Information files) the participating hospitals/institutions.

Response: Thank you for your suggestion. We have included the participating hospitals in Methods section under the sub-title “Setting and Study Design”

●The name of the colleague or the details of the professional service that edited your manuscript

●A copy of your manuscript showing your changes by either highlighting them or using track changes (uploaded as a *supporting information* file)

●A clean copy of the edited manuscript (uploaded as the new *manuscript* file)

Response: Thank you for your recommendation. However, it is difficult to use the recommended services, because the authors can’t afford the payment for the services. Instead, we have copy edited the whole manuscript for language usage, spelling and grammar with support from the communication experts at Wolaita Sodo University, consulting other researchers in the field, Wolaita Sodo University supported Office 365.

Response: We acknowledged your comment. We removed the retracted references and re-phrased the paragraphs containing retracted references.

Reviewers' comments:

Reviewer's Responses to Questions

Comments to the Author

1. Is the manuscript technically sounds, and do the data support the conclusions?

Reviewer #1: Yes

Reviewer #2: Yes

2. Has the statistical analysis been performed appropriately and rigorously?

Reviewer #1: Yes

Reviewer #2: Yes

3. Have the authors made all data underlying the findings in their manuscript fully available?

Reviewer #1: Yes

Reviewer #2: Yes

4. Is the manuscript presented in an intelligible fashion and written in standard English?

Reviewer #1: Yes

Reviewer #2: No

Response: We have corrected the typographical errors, the language usage, spelling and grammar with support from the communication experts at Wolaita Sodo University, consulting other researchers in the field, Wolaita Sodo University supported Office 365

5. Review Comments to the Author

Reviewer #1: Dear Authors

I would like to congratulate you for choosing an appropriate topic for research in healthcare setups. I have read complete manuscript. Please find below recommendations.

1) Add one separate section on “Lessons for International Healthcare Managers” at the end of manuscript.

Response: Thank you for your suggestion. We added the suggested section after the “Conclusion” section. 

2) Mention type of hospitals (private or public) for 03 selected hospitals in abstract and other sections of manuscript.

Response: Thank you for your suggestion. We have mentioned the participating hospitals in “abstract and methods” sections. 

3) Add one separate section on “Overall Healthcare Facilities in Ethiopia”, as a sub part of introduction section, which may include,

a. number of private and public healthcare service providers in the country

b. general comparison of quality of services between private and public service providers

Response: Thank you for your suggestion. We added the suggested section as part of our introduction. However, there were no officially published or clear report that may indicate the number of private healthcare facilities at national or local level in Ethiopia. Instead we discussed the available evidence on the specific number of public hospitals in Ethiopia. 

Regards,

(Reviewer)

Reviewer #2: 1. Summary of the research

The study title is: “Satisfaction with Regular Hospital Foodservices and Associated Factors among Adult Patients in Wolaita zone, Ethiopia: a facility-based cross-section study”. The study was conducted in 3 hospitals and established that about a third (33.6%) of the patients were satisfied with hospital food services. Factors such as the patients’ residence, monthly income, flavor of the meal, and ease of chewing the food were associated with satisfaction with regular Hospital Foodservices.

This is a unique study, because there are limited studies of this nature in sub-Sahara Africa, and this has implications on the nutrition status of hospitalized patients, as well as the negative consequences of malnutrition in hospital, such as length of stay, mortality, and cost of treatment.

2. Examples and evidence

a. MAJOR ISSUES

• The authors can consider working with an editor to improve the quality of the manuscript. There are several grammatical and sentence structure issues throughout the document that should be addressed – a few examples include lines 50-51, 55-56, 88-89, 173-174, 180-181.

Response: Thank you for your comment. We have corrected the raised issues in the main manuscript.

Consider paying closer attention to Lines 181-187. The information conveyed is not clear due to the language, yet it is a key finding in the study that needs clear and concise communication. See highlighted area here …..“Accordingly, 417(98.6%) and 386(91.3%) of the patients reported that they were receiving three meals daily and chewable meals respectively. Study participants had unacceptable experiences on majority of the food characteristics they were asked. In relation to this, all (100%) of the participants reported that there was no provision of water with their meal, and 399 (94.3%) of patients say that the flavor of the meal was not enough to eat. In addition, significant portion of participants, 230(54.4%) and 155(36.6%) had avoided the meal because of taste aversion and had skip the meal because of new meal type respectively,……”

Response: We acknowledged the comment. We have corrected the raised issues by re-phrasing paragraphs in the main manuscript.

• Indicate the steps taken to ensure the validity of the data collection instruments as well as procedures. For example, what was the source of the questionnaires used?

Response: Thank you for your comment. We have revised to make clear the data collection related issues under the sub-section ‘Data collection and quality assurance’. It was tried to ensure validity of the tool as follows: “The tool for data collection was developed after reviewing literature (18-23, 33-35). The tool was developed in English then translated to local languages (Amharic and Wolaita Donna) and back translated to English with different experts. The tool was pre-tested on 5% of study participants at another hospital which was not selected for the actual data collection. In order to maintain the internal consistency of the tool and to determine the reliability of the test, Cronbach’s alpha was applied. If the alpha was high (≥0.70), the item is considered to be reliable and the test is internally consistent”. 

• Indicate how the cut-off for the “overall patient (dis)satisfaction towards hospital foodservices” was determined (table 3). How was it decided that 142 were Satisfied, while 281 were Dissatisfied 281?

Response: We have tried to make clear concerning the cut-off point for satisfaction level under the sub-sections “Measurements and Patient’s satisfaction level with regular hospital foodservices”.

• Consider rephrasing the lines 214-217, by using interpretive language for the readers to get the results in a clearer manner, as this is a key component of the study. For instance, rephrase the section on meal flavor, easily chewable food to something like, …. “those who reported that they liked the flavor of the food were 2.6 times more likely to be satisfied compared to those who reported that they did not like the flavor …”.

THIS IS DONE IN THE DISCUSSION SECTION. Consider moving some of the content from the discussion section to the findings, so that the discussion is focused on situating this study in existing literature, as well as the implication of the findings.

Response: Thank you for your guidance. We have corrected it in the manuscript. 

• Consider merging contents of table 2 and table 3 that are presenting the same aspects of service. Currently, there is some overlap of information in the two tables. ONE SUGGESTION IS TO MAKE TABLE 2 AS CHARACTERISTICS OF SERVICE that were objectively determined e.g. number of meals, availability of water, availability of meal choice, whether they ate or skipped meals, among others. THEN IN TABLE 3, report all the subjective aspects that are the patients’ perceptions/ experiences.

Response: Thank you for your guidance. We have accepted the comment and corrected it in the manuscript. We have removed [Table 2] because the objectively determined services you suggested all are available in [Fig. 1] 

• Consider giving a general description of the foodservices so that the reader has some context, e.g. cover things like whether there was a variety of foods to choose from?

Response: We have described about the food variety of hospitals in a day in the main manuscript under sub-sections “Characteristics of foodservices in hospitals and Patient’s satisfaction level with regular hospital foodservices”. 

• Consider doing analysis by hospital as a factor since the different hospitals may have different characteristics and experiences

Response: Thank you for your suggestion. We have tried to have admission hospital as factor but it had not made association in our binary logistic regression, the p-value was 0.359 while our plan was to have variables with p-value <0.25 as candidate variables for multivariable logistic regression analysis [see our plan under sub-section ‘Data analysis’]. 

b. MINOR ISSUES

• In the introduction, examples from similar settings, particularly African countries, can be included to allow for comparison for levels of dissatisfaction (lines 58 and 67).

Response: Thank you for your suggestion. We have included level of dissatisfaction among hospital clients in African countries

• The introduction can be boosted to include factors associated with (dis)satisfaction now that this is part of the title of the study

Response: Thank you for your suggestion. We have included level of dissatisfaction among hospital clients in African countries

• Provide/cite the references used for the population and sampling content

Response: Thank you for your suggestion. We have modified the content and included references.

• Lines 86-83 can be revised to describe the sampling process in a step-by-step clearer language.

Response: Thank you for your suggestion. We have made it clear by re-phrasing paragraphs.

• Give a better description of the hospitals that were included in the study

Response: Thank you for your comment. We have included descriptions of the selected hospitals for the study.

• Put ETB in full first before abbreviating, in lines 176-177

Response: Thank you for your comment. We have corrected accordingly.

• Table 1: Indicate the equivalence of Ethiopian Birr to USD for international readers

Response: Thank you for your comment. We have corrected accordingly.

• Line 199, the meaning of “means of hygiene (95.7%)” is unclear, consider rephrasing

Response: Thank you for your suggestion. We have re-phrased as “Hand washing facility”.

• Correct the spelling for “TASTE” from “TEST” in Figure 1 and in Table 3, row 2

Response: Thank you for your comment. We have corrected accordingly.

• If information on patients’ satisfaction with food texture and/ or temperature is available, this can be added to the findings

Response: Thank you for your comment. There was information on the patient perception of meal temperature that asked patients to rate the warmness of the served meal under the sub-section “Patient’s satisfaction level with regular hospital foodservices”. However, there was no data concerning the food texture.

• The authors can ensure consistency in the way concepts are presented, by avoiding the interchanging of words that can result in different meanings in the text and tables. For instance, clarify whether when you are speaking with the patients on the chewability of the food (lines 182, 215, Table 4, Figure 1), was it that the food itself was tough to chew due to undercooking, or was it that the patients themselves had chewing issues, or both? The table 4 presentation versus that in the text (line 215) may differ in meaning

Response: Thank you for your suggestion. We have re-phrased as “well-cooked to chew easily”. Participants were asked to indicate whether the served meals were well-cooked to chew easily or tough due to undercooking. 

• Clarify meaning of “means of hygiene” in line 193

Response: Thank you for your suggestion. We have re-phrased as “Hand washing facility”.

6. PLOS authors have the option to publish the peer review history of their article (what does this mean?). If published, this will include your full peer review and any attached files.

Do you want your identity to be public for this peer review? For information about this choice, including consent withdrawal, please see our Privacy Policy.

Reviewer #1: Yes: Dr. Vishal Kamra

Reviewer #2: Yes: Irene Ogada

---

## [Editor Report · Decision Letter 1]

23 Jul 2021

PONE-D-20-36652R1

Satisfaction with Regular Hospital Foodservices and Associated Factors among Adult Patients in Wolaita zone, Ethiopia: a facility-based cross-sectional study

PLOS ONE

Dear Dr. Wakgari Binu Daga,

Thank you for submitting your manuscript to PLOS ONE. After careful consideration, we feel that it has merit but does not fully meet PLOS ONE’s publication criteria as it currently stands. Therefore, we invite you to submit a revised version of the manuscript that addresses the points raised during the review process.

1. Include some recommendations for policy and practice based on your findings.

2. Thoroughly copyedit your manuscript for language usage, spelling, and grammar. Rephrase the following lines to address this: Line: 51, 53, line 62 (Spelling of Malaysia), 62, 75, 86 (evidences, remove the "s"), 103 & 107 (put in articles and prepositions), 114, 157, 158, 163 (spelling of respondents), 182, 198 (participant's'), rephrase 208, 302, 304 for clarity, 259 grammar. 

3. Revise all aspects of methods and findings to past tense, since there is a lot of mixing of past and present tense. For example: lines 113, 159, 167, 211

4. Line 53 indicates that "Hospital malnutrition is the main problem". This should be revised since this may not be factually correct. There could be other problems, thus a neutral sentence like "Malnutrition is a problem ...."

5. Do you have studies that report associated factors such as patient characteristics like gender, income, residence? These can be included in the background. 

6. Indicate reference/ literature source for line 109, 

7. Address spacing between words, particularly after commas, throughout the document

8. Indicate the source of the 14 question items in the measurement section and how the cutoff of 32.6 out 138 of the total 70 points, was determined as appropriate. 

9. Clarify sentence 226 - does cleanliness of dishes mean the "food" of "plates/ utensils"

We look forward to receiving your revised manuscript.

Kind regards,

Irene Awuor Ogada

Academic Editor

PLOS ONE
---

## [Author Response · Author response to Decision Letter 1]

2 Sep 2021

PONE-D-20-36652R1

Satisfaction with Regular Hospital Foodservices and Associated Factors among Adult Patients in Wolaita zone, Ethiopia: a facility-based cross-sectional study

PLOS ONE

Dear Dr. Wakgari Binu Daga,

Thank you for submitting your manuscript to PLOS ONE. After careful consideration, we feel that it has merit but does not fully meet PLOS ONE’s publication criteria as it currently stands. Therefore, we invite you to submit a revised version of the manuscript that addresses the points raised during the review process.

1. Include some recommendations for policy and practice based on your findings.

Response: We acknowledged your comment and addressed the raised issue in the manuscript

2. Thoroughly copyedit your manuscript for language usage, spelling, and grammar. Rephrase the following lines to address this: Line: 51, 53, line 62 (Spelling of Malaysia), 62, 75, 86 (evidences, remove the "s"), 103 & 107 (put in articles and prepositions), 114, 157, 158, 163 (spelling of respondents), 182, 198 (participant's'), rephrase 208, 302, 304 for clarity, 259 grammar. 

Response: We acknowledged the comment and addressed the raised issue in the manuscript

3. Revise all aspects of methods and findings to past tense, since there is a lot of mixing of past and present tense. For example: lines 113, 159, 167, 211

Response: We acknowledged the comment and addressed the raised issue in the manuscript

4. Line 53 indicates that "Hospital malnutrition is the main problem". This should be revised since this may not be factually correct. There could be other problems, thus a neutral sentence like "Malnutrition is a problem ...."

Response: we acknowledged your comment and we have rephrased the sentence

5. Do you have studies that report associated factors such as patient characteristics like gender, income, residence? These can be included in the background. 

Response: we acknowledged your comment. As to the knowledge of the Authors, evidence identified low monthly income as independent associated factor from the patient characteristics. This is mentioned in background of the manuscript. 

6. Indicate reference/literature source for line 109,

Response: thank you for your advice. Reference is indicated in the manuscript as “Wolaita Zonal Health Department fiscal year report, 2019” immediately after the sentence. 

7. Address spacing between words, particularly after commas, throughout the document

Response: Thank you for your comment and we have addressed the raised issue throughout the manuscript

8. Indicate the source of the 14 question items in the measurement section and how the cutoff of 32.6 out 138 of the total 70 points, was determined as appropriate. 

Response: We acknowledged the comment and we have recognized that there is typo-error on the figure assigned as the cut-off point. We have rephrased as follows. […….To see the total score of each respondent, the points obtained from the 14 items were computed. A total score ranges from 14 to70 points for each respondent in which higher scores indicate greater satisfaction. A mean score of the likert scale was computed giving 2.44. A client was regarded as satisfied (if scored ≥2.44); otherwise, the client was considered as dissatisfied (mean score of <2.44)].

9. Clarify sentence 226 - does cleanliness of dishes mean the "food" of "plates/ utensils"

Response: Thank you for your comment and we have addressed the raised issue by rephrasing paragraphs in the manuscript

If applicable, we recommend that you deposit your laboratory protocols in protocols.io to enhance the reproducibility of your results. Protocols.io assigns your protocol its own identifier (DOI) so that it can be cited independently in the future. For instructions, see: http://journals.plos.org/plosone/s/submission-guidelines#loc-laboratory-protocols. Additionally, PLOS ONE offers an option for publishing peer-reviewed Lab Protocol articles, which describe protocols hosted on protocols.io. Read more information on sharing protocols at https://plos.org/protocols?utm_medium=editorial-email&utm_source=authorletters&utm_campaign=protocols.

We look forward to receiving your revised manuscript.

Kind regards,

Irene Awuor Ogada

Academic Editor

PLOS ONE

Journal Requirements:

Response: we have edited the reference #23 “Mohamed A. Al-Torky EAM, Fouad MA.Yousef, Nesreen AM. Ali. Inpatients’ satisfaction with food services in Sohag University Hospital. The Egyptian Journal of Community Medicine 2016;34(2)”. It is edited as “Mohamed A. Al-Torky EAM, Fouad MA.Yousef, Nesreen AM. Ali. Inpatients’ satisfaction with food services in Sohag University Hospital. The Egyptian Journal of Community Medicine, 2016; 34(2): 33-45. https://doi.org/10.21608/ejcm.2016.651”

Response: We have checked ‘Figure 1: Characteristics of hospital foods among patients in hospitals, Wolaita zone, Ethiopia 2019’ for PLOS requirements using the PACE tool

---

## [Decision Letter · Decision Letter 2]

7 Feb 2022

Satisfaction with Regular Hospital Foodservices and Associated Factors among Adult Patients in Wolaita zone, Ethiopia: a facility-based cross-sectional study

PONE-D-20-36652R2

Dear Dr. Daga,

We’re pleased to inform you that your manuscript has been judged scientifically suitable for publication and will be formally accepted for publication once it meets all outstanding technical requirements.

Kind regards,

Sergio A. Useche, Ph.D.

Academic Editor

PLOS ONE

Additional Editor Comments (optional):

Reviewers' comments:

Reviewer's Responses to Questions

**Comments to the Author**

1. If the authors have adequately addressed your comments raised in a previous round of review and you feel that this manuscript is now acceptable for publication, you may indicate that here to bypass the “Comments to the Author” section, enter your conflict of interest statement in the “Confidential to Editor” section, and submit your "Accept" recommendation.

Reviewer #1: All comments have been addressed

2. Is the manuscript technically sound, and do the data support the conclusions?

Reviewer #1: Yes

3. Has the statistical analysis been performed appropriately and rigorously? 

Reviewer #1: Yes

4. Have the authors made all data underlying the findings in their manuscript fully available?

Reviewer #1: Yes

5. Is the manuscript presented in an intelligible fashion and written in standard English?

Reviewer #1: Yes

6. Review Comments to the Author

Reviewer #1: Dear Authors

Congratulations !!!

All comments are addressed Successfully.

Best Wishes,

(Reviewer)

7. PLOS authors have the option to publish the peer review history of their article (what does this mean?). If published, this will include your full peer review and any attached files.

Reviewer #1: **Yes: **Dr. Vishal Kamra

---

## [Editor Report · Acceptance letter]

9 Feb 2022

PONE-D-20-36652R2 

Satisfaction with Regular Hospital Foodservices and Associated Factors among Adult Patients in Wolaita zone, Ethiopia: a facility-based cross-sectional study 

Dear Dr. Daga:

I'm pleased to inform you that your manuscript has been deemed suitable for publication in PLOS ONE. Congratulations! Your manuscript is now with our production department. 

Kind regards, 

on behalf of

Dr. Sergio A. Useche 

Academic Editor

PLOS ONE